# Flavivirus maturation leads to the formation of an occupied lipid pocket in the surface glycoproteins

Max Renner [1,6], Wanwisa Dejnirattisai[2,6], Loïc Carrique [1], Itziar Serna Martin[3], Dimple Karia[1], Serban L. Ilca [1], Shu F. Ho[1], Abhay Kotecha [1], Jeremy R. Keown [1], Juthathip Mongkolsapaya [2,4✉], Gavin R. Screaton [2✉] & Jonathan M. Grimes [1,5✉]

Flaviviruses such as Dengue (DENV) or Zika virus (ZIKV) assemble into an immature form within the endoplasmatic reticulum (ER), and are then processed by furin protease in the trans-Golgi. To better grasp maturation, we carry out cryo-EM reconstructions of immature Spondweni virus (SPOV), a human flavivirus of the same serogroup as ZIKV. By employing asymmetric localised reconstruction we push the resolution to 3.8 Å, enabling us to refine an atomic model which includes the crucial furin protease recognition site and a conserved Histidine pH-sensor. For direct comparison, we also solve structures of the mature forms of SPONV and DENV to 2.6 Å and 3.1 Å, respectively. We identify an ordered lipid that is present in only the mature forms of ZIKV, SPOV, and DENV and can bind as a consequence of rearranging amphipathic stem-helices of E during maturation. We propose a structural role for the pocket and suggest it stabilizes mature E.

[1] Division of Structural Biology, The Wellcome Centre for Human Genetics, University of Oxford, Oxford, UK. [2] Nuffield Department of Medicine, The Wellcome Centre for Human Genetics, University of Oxford, Oxford, UK. [3] Bijvoet Centre for Biomolecular Research, Department of Chemistry, Faculty of Science, Utrecht University, Utrecht, The Netherlands. [4] Dengue Hemorrhagic Fever Research Unit, Office for Research and Development, Faculty of Medicine Siriraj Hospital, Mahidol University, Bangkok, Thailand. [5] Science Division, Diamond Light Source Ltd, Didcot, UK. [6] These authors contributed equally: Max Renner, Wanwisa Dejnirattisai. ✉email: juthathip.mongkolsapaya@well.ox.ac.uk; gavin.screaton@medsci.ox.ac.uk; jonathan@strubi.ox.ac.uk

The genus *Flavivirus* contains many important mosquito-borne human pathogens, such as dengue virus (DENV) and Zika virus (ZIKV). Flaviviruses represent a significant economic and health-care burden to affected countries, particularly in Southeast Asia and South America[1]. The surface of mature, infectious flaviviruses is composed of an icosahedral shell of 90 flat-lying envelope (E) protein dimers[2–4], the main target of humoral immunity[5–7]. E contains the fusion peptide, which is responsible for insertion into host membranes during infection[8]. Within infected host cells, progeny immature viruses bud into the lumen of the endoplasmatic reticulum (ER) and follow the secretory pathway through the Golgi apparatus[3]. In contrast to their mature counterparts, immature virions possess 60 protruding spike-like trimers of precursor membrane (prM)-E complexes (prM$_3$E$_3$)[9,10]. The pr domain of prM binds to E, obscuring the fusion peptide and thereby preventing unproductive fusion within the already infected cell[11]. The M domain of prM is embedded in the membrane and connected to pr via a flexible linker. When immature virions reach the low-pH environment of the trans-Golgi network (TGN), the trimeric prM$_3$E$_3$ spikes rearrange into the flat-lying conformation, characteristic of the mature virus[12,13]. The TGN-resident protease furin[14] then cleaves a recognition site between pr and M. pr remains associated with E, inhibiting its fusion function, until the virus is transported outside of the cell. There, the neutral pH environment triggers the dissociation of E and pr, rendering the virus infectious[12].

Structural studies of immature flaviviruses via cryogenic electron microscopy (cryo-EM) have thus far been limited to low to medium resolutions[9,10,13,15,16], hampering our mechanistic understanding of the maturation process. Here we investigated flavivirus maturation by high-resolution cryo-EM. We chose to carry out this study using Spondweni virus (SPOV)[17], a human flavivirus that is closely related to ZIKV (~75% amino-acid identity) and has remained structurally uncharacterized until now. Owing to their similarity, SPOV was initially misclassified as a strain of ZIKV in neutralization tests[18,19]. SPOV was identified in Sub-Saharan Africa, but has recently been detected in *Culex quinquefasciatus* mosquitoes in Haiti, indicating that the virus may be adapting to vectors that prefer human hosts and that the epidemiology may therefore be changing[20,21]. In addition, mutual infection enhancement between DENV, ZIKV, and SPOV by immune sera has been demonstrated in cell culture[22]. We have used localized reconstruction[23] and focused refinement to obtain a 3.8 Å resolution cryo-EM structure. In contrast to previous cryo-EM studies, the improved resolution enabled the building of a refined atomic model for (prM)-E, including the crucial furin protease recognition site. For direct comparison, we have also carried out cryo-EM reconstructions of mature SPOV (2.6 Å resolution), and DENV (3.1 Å resolution). Our data resolve how the furin recognition site on pr interacts with neighboring E and thus is sequestered in the trimeric spike and how prM is anchored to E via a strictly conserved and mutation-sensitive histidine. Comparison with mature virus reveals that rearrangement of amphipathic helices in E during maturation leads to the formation of a binding pocket for a lipid headgroup, which is present in ZIKV, SPOV, and DENV. We propose the lipid stabilizes the conformation of membrane-associated helices of E and the binding site may be a potential target for the development of antiviral compounds. Based on our structures, we suggest a function for the immature trimeric state of flavivirus glycoproteins, and a pathway for glycoprotein rearrangement during maturation.

## Results

To facilitate the collection of high-quality cryo-EM data outside of biosafety containment, we developed an optimized UV-exposure protocol that completely inactivates virus, while fully maintaining sample integrity (Supplementary Fig. 1a, b). Our samples contained a mixture of mature and immature particles (Supplementary Fig. 1b, black and white arrows, respectively). By only selecting immature particles we were able to obtain a reconstruction at 7.8 Å resolution (Fig. 1a, b and Table 1). The map shows the overall architecture of the spike-like trimers and extended areas of exposed membrane around the threefold and fivefold vertices. The virion possesses a diameter of 560 Å, comparable to that of ZIKV and DENV[9,10].

At 7.8 Å resolution, the interpretability of a map is limited. We used localized reconstruction[23] and focused refinement to provide high-resolution insight into prM-E complexes (Supplementary Fig. 2). We obtained a 4.2 Å resolution map of the prM$_3$E$_3$ trimer (Supplementary Fig. 3) and 3.8 Å resolution for monomeric prM$_1$E$_1$ composing the trimeric spikes (Fig. 1c, d and Supplementary Movie 1). In the structure, the pr domain sits on top of the E protein, covering the fusion loop and burying a combined total surface area of ~3100 Å$^2$. Within the context of the spike, three copies of prM$_1$E$_1$ form an asymmetric trimer (i.e., not following C3-symmetry), in which the three pr domains sitting at the tips of E come together and interact (Fig. 1d and Supplementary Fig. 4). The RSRR furin recognition motif, which is critical for maturation, is found towards the end of the pr domain (Fig. 1c, e). Polar interactions between E residues (in particular Glu62 and Glu245) and positively charged residues on pr fix the position of the furin site (Fig. 1e). Furthermore, hydrophobic prM residues (Ile96 and Leu98) directly downstream of the furin site, bury into the surface of E, providing additional stabilization.

A central question in flavivirus maturation is how low pH in the trans-Golgi induces the necessary conformational rearrangements in the prM$_3$E$_3$ spike, that lead to exposure of the furin site on pr. E and prM are anchored in the viral membrane via their respective membrane-associated regions. In E, these encompass two amphipathic (which lie flat on the membrane) and two transmembrane (TM) helices (Fig. 1d), whereas M possesses one amphipathic and two TM helices (Fig. 2a). The pr domain of prM connects to the membrane-associated M domain via a linker region encompassing residues 102–119. No density is observable for this region, possibly indicating high conformational dynamics. However, the last clearly visible residue prior to this flexible region is a histidine (His101), which is inserted in a hydrophobic pocket on E (Fig. 2a, b).

Although the prM sequence is relatively diverse compared with that of E, His101 is highly conserved (Fig. 2c). A decrease in pH would lead to protonation of His101, destabilizing this residue in the hydrophobic pocket and promoting prM-E dissociation. These structural data are underlined by previous functional studies, which identified the equivalent histidines in Japanese Encephalitis virus (JEV)[24] and DENV[25] to be important for the low-pH induced dissociation of prM and E. One study showed that mutation of the histidine to alanine inhibited the processing of prM[25]. The mutant could be rescued by a charge-inversion (Glu to Lys mutation) of an E protein residue, which in our structure is responsible for sequestering the furin site (Glu62 in Fig. 1e). Thus, destabilization of the upstream furin site could counteract mutation of the pH-switch. These data support His101, and the equivalent histidines in other flaviviruses, as important pH sensors during maturation and to anchor the prM linker.

From the same data set, we solved the structure of the mature form of SPOV (which lacks the pr domain), allowing us to examine which changes occur during maturation. We selected mature particles and reconstructed a map to 2.6 Å resolution (Fig. 3a and Supplementary Fig. 5). The virion has a smooth

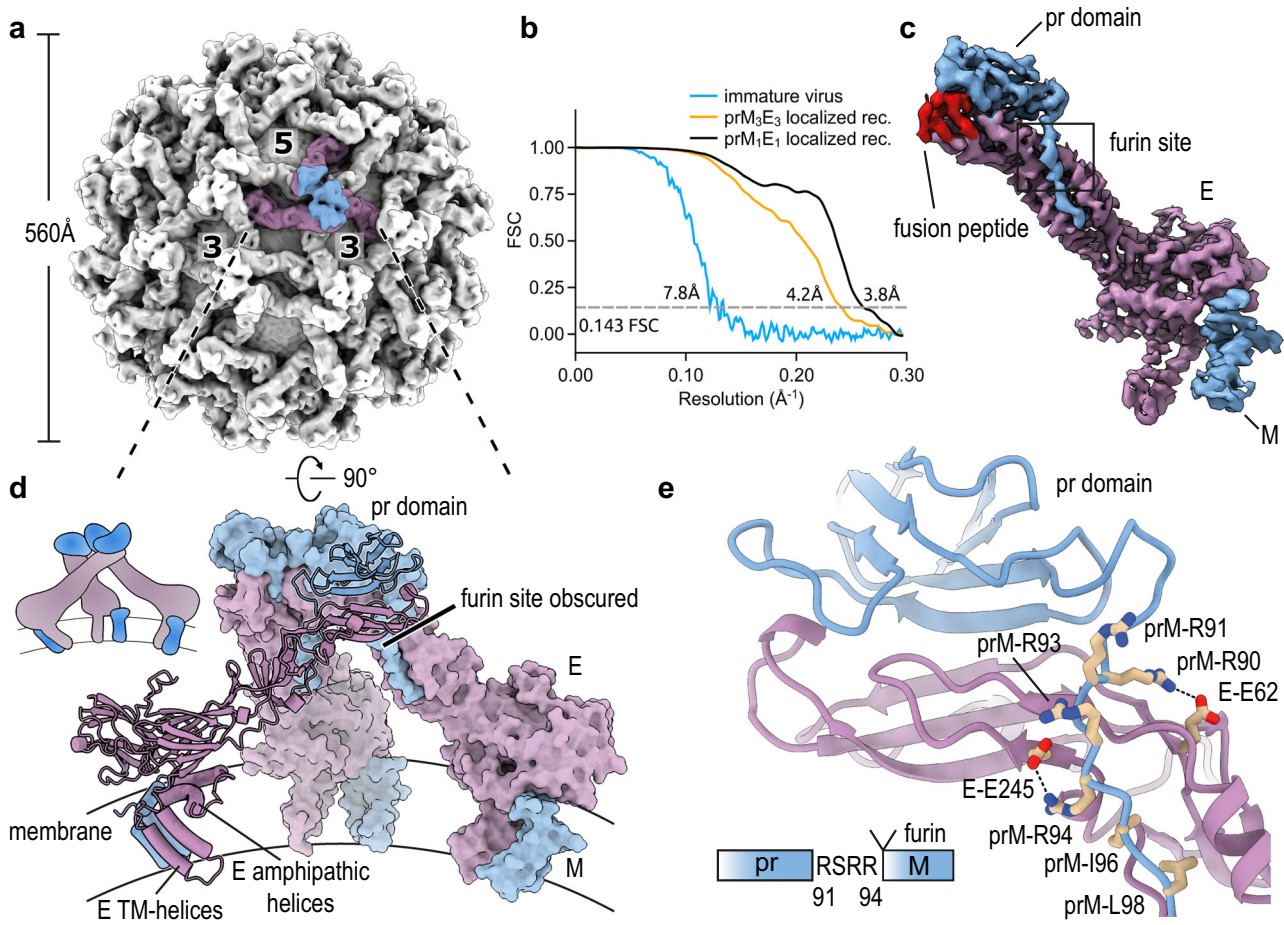

**Fig. 1 Cryo-EM structures of immature SPOV. a** Cryo-EM density map of immature SPOV (icosahedral symmetry applied). Bold numbers indicate threefold and fivefold vertices of the virion. The prM₃E₃ trimer of one asymmetric unit is colored (prM in blue, E in purple). Approximate dimensions of the virion are indicated by the accompanying scale bar. **b** Fourier shell correlation (FSC) plots of reconstructions using gold-standard refinement in RELION. Approximate map resolutions according to the 0.143 FSC cutoff are indicated for all reconstructions. Curves are shown for masked maps. **c** Cryo-EM density map of the prM₁E₁ complex at 3.8 Å resolution, obtained via localized reconstruction and focused refinement. The map served as the basis for building an initial atomic model. E is colored in purple, prM in blue. Locations of the furin protease recognition motif and fusion peptide (colored in red) are indicated. **d** Atomic model of the prM₃E₃ trimeric spike, refined against the 4.2 Å resolution map (orientation rotated by 90° relative to **a**). One copy of prM1E1 is shown in ribbon representation, with transmembrane (TM) helices indicated, and the two others as surfaces. The viral membrane is shown schematically. A cartoon depicting the organization of the trimer is shown for clarity. **e** Close-up of the area surrounding the furin recognition motif as indicated in **c**. E and prM residues involved in stabilizing the site are labeled. A schematic is included at the bottom of the panel showing the sequence after which furin cleaves pr and M.

appearance, typical for members of the genus *Flavivirus*, with a diameter of ~460 Å. The E proteins on the surface are arranged as antiparallel dimers (Fig. 3b), forming a tight cage that almost entirely covers the lipid bilayer. Unlike DENV (which is also glycosylated at Asn67), SPOV E possesses only a single glycosylation site at position Asn154 (Fig. 3b, c). The density indicated core fucosylation at the first GlcNAc, which is consistent with previous studies showing that fucosylation is present at the Asn153-glycan (DENV numbering) but not the Asn67-glycan for all DENV serotypes[26]. In ZIKV, the Asn154-glycan has been shown to be important for pathogenicity[27], and for oral infectivity in *A. aegypti* mosquitoes[28]. The lipid membrane shows a kinked morphology and bilayer thinning[29], induced by the membrane-associated helices of the embedded proteins (Supplementary Fig. 5a).

We then assessed the conformational changes within the ectodomain of E in the presence and absence of the pr domain. Despite the dramatically differing overall morphologies of mature and immature virions, the conformations of the E ectodomains are remarkably similar in both states. We do observe a downward

hinge movement of ~15° in mature E (Supplementary Fig. 5c). Such hinge movements are typical of E proteins and have been observed throughout its various structural states described in the literature[30]. We also find that a β-hairpin (residues 244–257) of E, located at the interface to the pr domain, shifts its position in the immature virus to optimally bind the pr domain (Fig. 3d, immature E hairpin in gray). Specifically, polar interactions between the E hairpin and Asp64/Asp66/Asp40 of pr stabilize binding of the pr domain to E. His250 is clearly visible at the interface, and deprotonation of this conserved residue in DENV has been suggested to be involved in the dissociation of cleaved pr at neutral pH[31].

Although the domain architecture of E remains constant, there is a significant reorganization of membrane-associated helices of E, going from the immature to the mature form (Fig. 4a, b). In mature E, the amphipathic helix H2 is splayed away from the transmembrane bundle of TM1 and TM2, whereas amphipathic helix H1 has reoriented by almost 180°, relative to the helix bundle (Fig. 4b). The position of the membrane-associated helices has also shifted relative to the ectodomain of E (Supplementary

**Table 1 Cryo-EM data collection, processing, and model refinement statistics.**

| | Immature SPOV virion PDB-ID: 6ZQW EMD-11372 | Immature SPOV trimeric spike (prM₃E₃) PDB-ID: 6ZQJ EMD-11366 | Immature SPOV heterodimer (prM₁E₁) PDB-ID: 6ZQI EMD-11364 | Mature SPOV virion PDB-ID: 6ZQV EMD-11371 | Mature DENV2 virion PDB-ID: 6ZQU EMD-11370 |
|---|---|---|---|---|---|
| *Data collection and processing* | | | | | |
| Microscope | Titan Krios | Titan Krios | Titan Krios | Titan Krios | F30 Polara |
| Detector | Gatan K3 | Gatan K3 | Gatan K3 | Gatan K3 | Gatan K2 Summit |
| Voltage (kV) | 300 | 300 | 300 | 300 | 300 |
| Recording mode | Super resolution | Super resolution | Super resolution | Super resolution | Super resolution |
| Electron exposure (e⁻/Å²) | 30.00 | 30.00 | 30.00 | 30.00 | 32.00 |
| Defocus range (μm) | −0.5 to −2.0 | −0.5 to −2.0 | −0.5 to −2.0 | −0.5 to −2.0 | −0.7 to −2.7 |
| Magnification | 105,000 | 105,000 | 105,000 | 105,000 | 59,000 |
| Movie pixel size (Å) | 0.4225 | 0.4225 | 0.4225 | 0.4225 | 0.44 |
| Final map pixel size (Å) | 1.69 | 1.69 | 1.69 | 0.845 | 0.88 |
| Symmetry imposed | I1 | C1 | C1 | I1 | I1 |
| Initial particle images (no.) | 42,880 | 2,572,800 subparticles | 2,572,800 subparticles | 160,341 | 6676 |
| Final particle images (no.) | 11,768 | 305,017 subparticles | 281,619 subparticles | 63,222 | 2938 |
| Map resolution at 0.143 FSC threshold (Å) | 7.8 | 4.2 | 3.8 | 2.6 | 3.1 |
| Map sharpening *B* factor (Å²) | −300 | −200 | −130 | −30 | −50 |
| *Model refinement* | | | | | |
| Initial model used (PDB code) | – | 6ZQI (this study) | 3C5X | 6CO8 | 3J27 |
| FSC model vs. map at 0.5 threshold (Å) | – | 4.33 | 4.07 | 2.65 | 3.27 |
| CC model vs map (masked) | – | 0.75 | 0.78 | 0.89 | 0.79 |
| *Model composition* | | | | | |
| Non-hydrogen atoms | – | 15,096 | 5032 | 13,476 | 13,209 |
| Protein residues | – | 1965 | 655 | 1734 | 1,698 |
| Non-protein residues | – | 3 | 1 | 12 | 6 |
| *B factors (Å²)* | | | | | |
| Protein | – | 100.2 | 78.8 | 69.70 | 61.09 |
| Non-protein | – | 152.4 | 89.0 | 92.16 | 71.91 |
| *R.m.s deviations* | | | | | |
| Bond lengths (Å) | – | 0.002 | 0.002 | 0.006 | 0.002 |
| Bond angles (°) | – | 0.539 | 0.549 | 0.538 | 0.439 |
| *Validation* | | | | | |
| MolProbity score | – | 1.66 | 1.53 | 2.01 | 1.69 |
| Clashscore | – | 6.10 | 5.73 | 5.35 | 9.61 |
| Poor rotamers (%) | – | 1.55 | 1.12 | 4.39 | 1.39 |
| *Ramachandran plot* | | | | | |
| Favored | – | 96.97 | 96.92 | 96.46 | 97.36 |
| Allowed | – | 2.93 | 3.08 | 4.39 | 2.37 |
| Disallowed | – | 0.10 | 0.00 | 0.00 | 0.00 |

All SPOV reconstructions were carried out using a single data set.

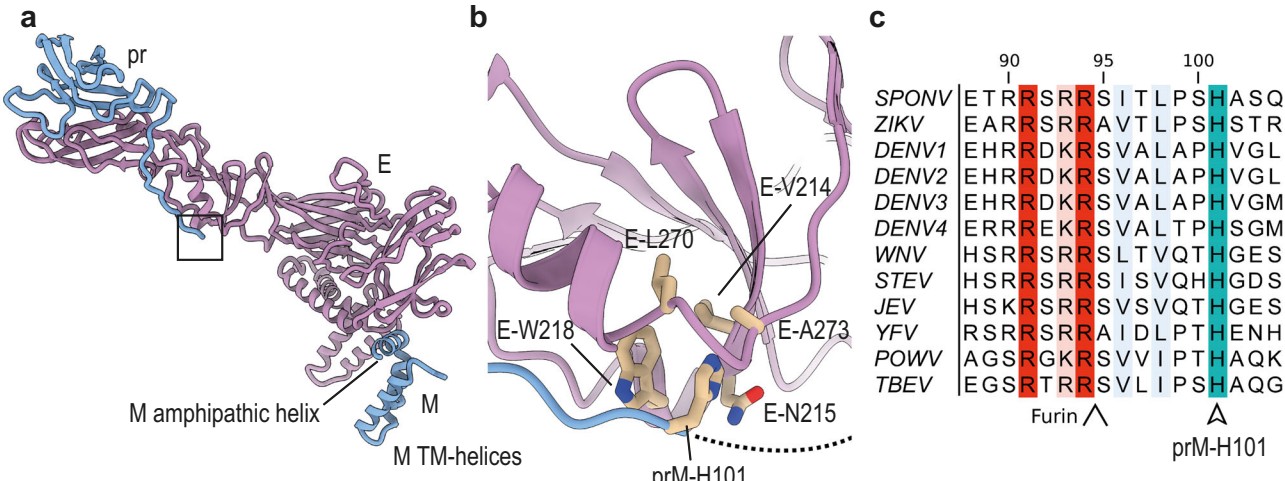

**Fig. 2 Anchoring of the prM linker to E. a** Zoomed out view of an E-prM complex. The boxed region indicates the location of the prM-H101 pocket. **b** Zoomed-in view of the boxed region indicated in **a**. A hydrophobic pocket on the surface of E is shown, into which prM-H101 is inserted. Involved E and prM residues are labeled. **c** Multiple sequence alignment (MSA) of flaviviral prM sequences. The positions of the conserved prM-H101 (SPOV numbering) are indicated, as well as the upstream furin site. *SPONV* Spondweni virus, *ZIKV* Zika virus, *DENV* Dengue virus, *WNV* West Nile virus, *SLEV* Saint Louis encephalitis virus, *JEV* Japanese encephalitis virus, *YFV* Yellow Fever virus, *POWV* Powassan virus, *TBEV* Tick-borne encephalitis virus.

Fig. 5c). Notably, although TM and amphipathic helices are tightly packed together in the immature virus, the reorganization has left a pocket-like gap in between the helices in mature SPOV. We examined the EM map in the vicinity of the pocket and found that it is occupied by strong density resembling the headgroup of a phosphatidylethanolamine lipid (PE), with partially resolved fatty-acid tails (Fig. 4b, c). PE is the most abundant phospholipid in the lipidome of *Aedes aegypti* and *Aedes albopictus* cells[32]. The PE fills the pocket between the membrane-associated helices of mature E, thereby keeping them at a fixed distance and position.

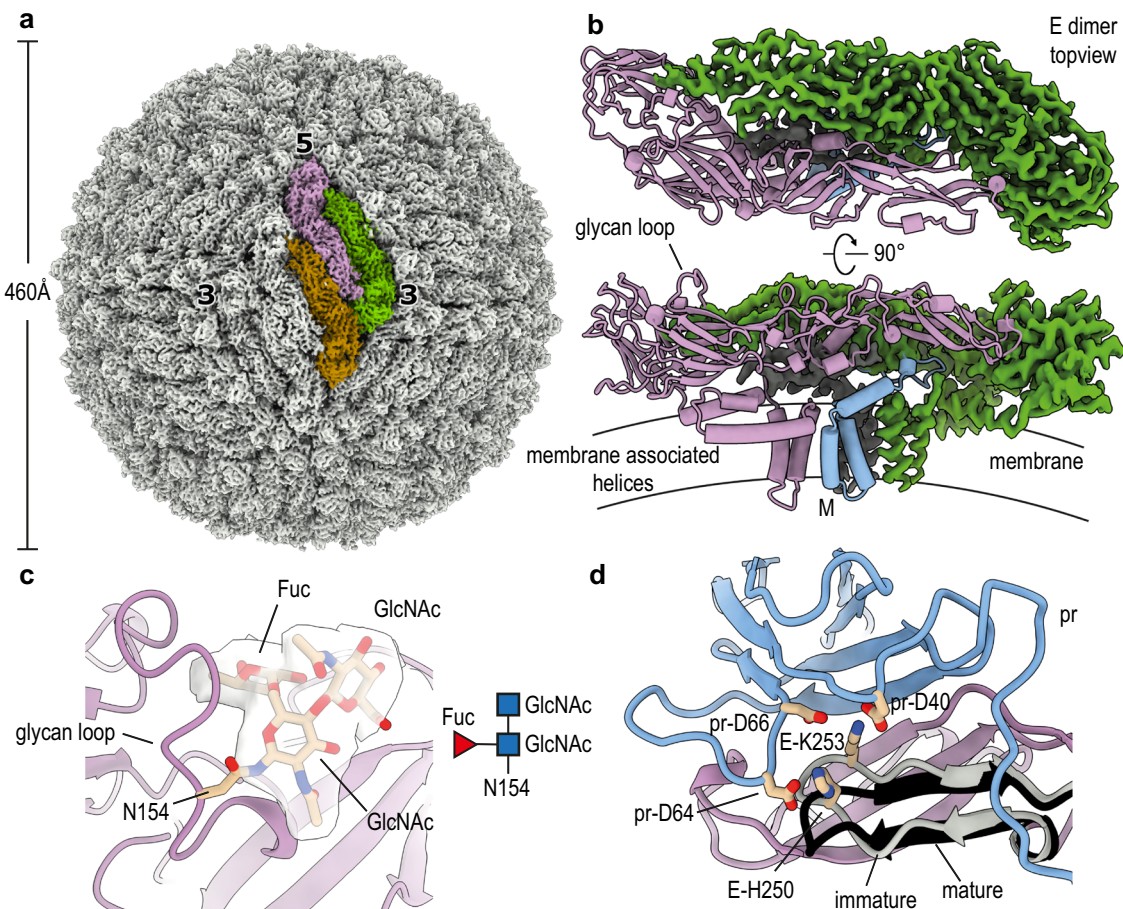

**Fig. 3 2.6 Å resolution cryo-EM structure of mature SPOV. a** Cryo-EM density map of mature SPOV. Bold numbers indicate threefold and fivefold vertices of the virion. Each icosahedral asymmetric unit (ASU) contains three copies of E (colored brown, purple, and green for one ASU) and three copies of M, concealed below the E proteins. Approximate dimensions of the virion are indicated by the accompanying scale bar. **b** Top-view and side-view of an antiparallel dimer of E, which compose the surface of the mature virus. One copy is depicted in ribbon representation (purple), whereas for the other copy the cryo-EM map is shown (green). The M protein is indicated in the side-view, as well as a schematic of the viral membrane. A surface-exposed loop containing the glycan site is labeled. **c** Close-up view showing the density map of the glycan linked to N154 of E. There was clear density for the two first N-acetylglucosamines (GlcNac) and core fucosylation (Fuc). **d** Superposition of the pr-binding interface of E in mature and immature states. The comparison shows that a hairpin close to the pr-binding site of E shifts its position to better engage pr (immature hairpin in gray, mature in black). Residues on the hairpin involved in binding pr are shown as sticks and labeled.

Hydrophobic residues surround the pocket (Phe454, Leu499) and a histidine (His447) is found packing against the lipid, in close vicinity to the phosphate group of PE (Fig. 4b). Multiple sequence alignment of flavivirus E sequences reveals that the histidine adjacent to the pocket is conserved across all analyzed sequences (Fig. 4d).

Considering the conservation, we wanted to assess if a lipid is present in related pathogenic flaviviruses, such as ZIKV or DENV. We inspected the density in this region in the deposited 3.1 Å map of mature ZIKV[4] (EMD-7543). Although the map is lower resolution than that of SPOV, we can observe prominent density for the lipid in the identical position in ZIKV (Fig. 4e). Subsequent to the release of the preprint version of our manuscript[33], another study also reported the presence of the lipid in ZIKV[34]. Next, we examined the previously solved cryo-EM structure of DENV2[2]. Because the available 3.6 Å resolution map did not resolve unambiguously the presence of a lipid, we carried out our own cryo-EM reconstruction of DENV2 (Supplementary Fig. 6). The reconstruction reached a resolution of 3.1 Å and again revealed a lipid in the same region (Fig. 4f), suggesting that the binding pocket is conserved in flaviviruses. Notably, a functionally equivalent observation has recently been made in Sindbis

alphavirus, where a lipid pocket factor was found buttressed in between the membrane-associated helices of the glycoproteins[35] (Fig. 4g). A histidine was also found packing against the pocket factor in this case, and has been suggested to be protonated at low pH during virus entry and to be involved in subsequent collapse of the lipid-binding pocket[35].

To further investigate the role of the conserved residues adjacent to the lipid, we constructed mutant viruses via Gibson assembly[36] and assessed their growth. We chose to carry out mutagenesis in DENV2, as this is the more amenable system and impactful pathogen. We found that no virus could be recovered after exchange of His437 (equivalent to His447 in SPOV, Fig. 4b) to either a negatively charged glutamate or a small and neutral alanine (Fig. 4h). Similarly, we observed no virus recovery upon introducing a bulky Tyr sidechain at position Gly441 (Gly451 in SPOV). This glycine is in close contact with the lipid and any other amino-acid besides Gly would sterically displace the lipid, which may explain why it is strictly conserved at this position (Fig. 4d). Although we cannot exclude with absolute certainty that these mutations interfere with the folding of E, the results suggest that an intact lipid-binding pocket is important in the virus lifecycle, with a specific role for the adjacent histidine.

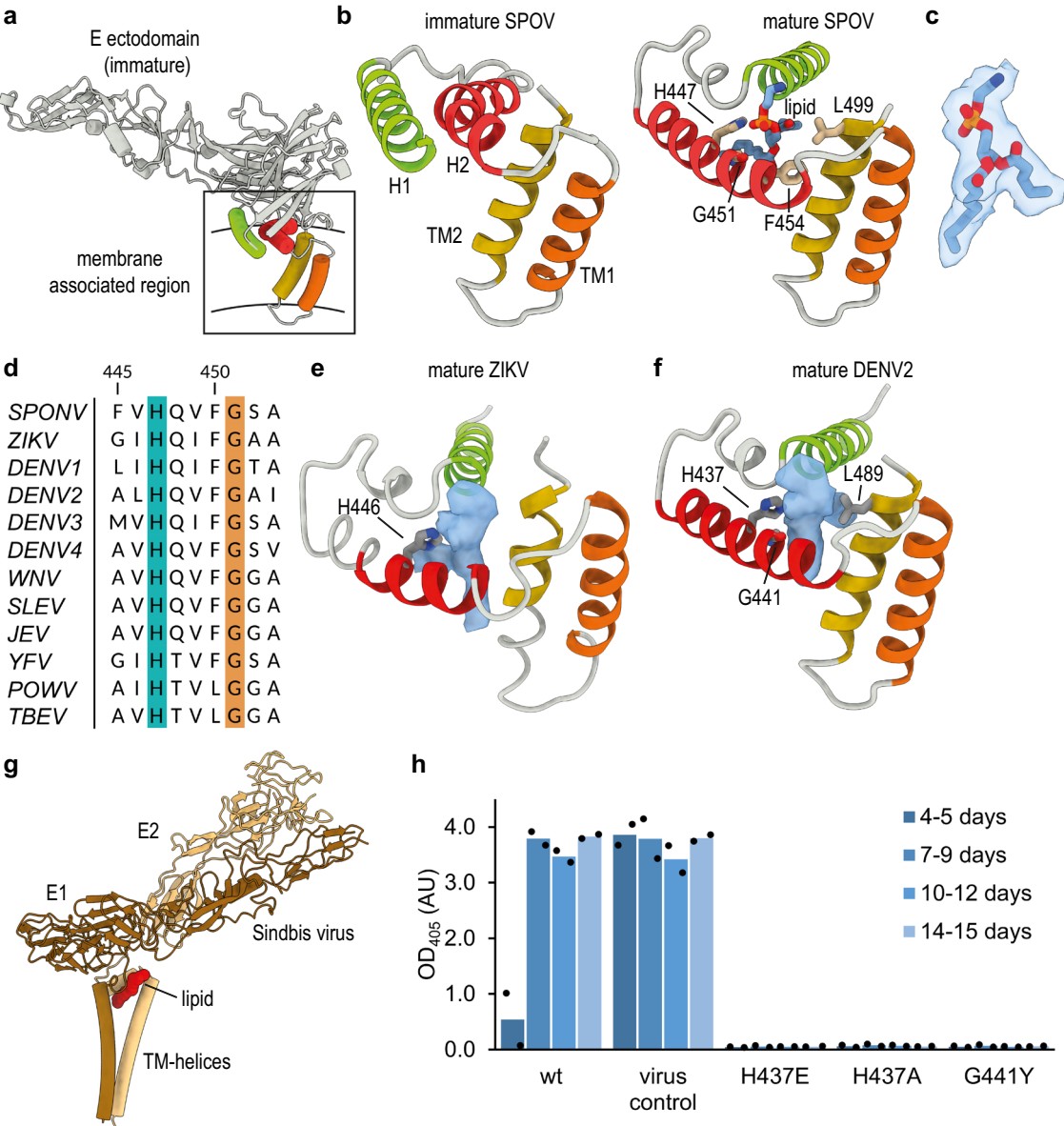

**Fig. 4 Formation of a lipid pocket by rearrangement of membrane-associated helices. a** Position and organization of membrane-associated helices in immature E. Two amphipathic helices lie flat on the membrane (colored green and red), whereas two transmembrane (TM) helices span it (colored yellow and orange). **b** Close-up view of the membrane-associated helices of immature E and mature E of SPOV. Amphipathic helices are numbered H1 and H2, TM helices TM1 and TM2. In mature E (on the right), the amphipathic helices have reorganized, opening up a pocket that is filled with density resembling a phosphatidylethanolamine (PE) lipid. The lipid is shown in stick representation. The density fit is shown separately in **c** for clarity. Residues in close vicinity are shown as sticks and labeled appropriately. **c** Lipid fitted into the cryo-EM map at the binding pocket. **d** MSA of E sequences located next to the lipid, highlighting the conservation of H447 and G451 (SPOV numbering). Virus abbreviations as in Fig. 2. **e** Density present in the pocket of mature ZIKV (accession code: EMDB-7543). **f** Density present in the pocket of mature DENV2 (this study). **g** Structure of Sindbis virus E1 and E2 (PDB-ID: 6IMM) illustrating the similarity of the position of a glycoprotein associated lipid (colored in red). **h** Virus recovery after Gibson assembly of mutant viruses. Wild-type and mutants of DENV2/16681 were constructed via Gibson assembly and propagated in C6/36 cells. Production of virus was assessed after given intervals by ELISA. Cells infected with virus stock served as positive control. The results of $n = 2$ biologically independent experiments are shown.

## Discussion

A remarkable reorganization of the flavivirus protein shell occurs during maturation and, with no structures of intermediates between the trimeric and dimeric arrangements of E available, the trajectory of this process remains undetermined. Previously, it has been speculated that large-scale rotations of the E ectodomains might be involved in reorienting immature glycoproteins into their mature positions. It has been argued that in the course of this process the prM linker acts as a "drawstring" and pulls on the ectodomains, thereby rotating them. However, simultaneous pivoting of multiple ectodomains in neighboring asymmetric units of the virion is, in our view, sterically problematic and the energetic driver of the pulling motion is unclear. Instead, we anticipate that simple glycoprotein translations in the fluid lipid bilayer impose a much lower energy barrier and are sterically feasible. We propose that translations of glycoproteins in the immature-mature transition make use of the available space in the membrane at the five- and threefold vertices (Fig. 1a) to reposition the proteins. Indeed, such translational fluctuations may explain the difficulties in achieving high resolutions in

cryo-EM of immature flaviviruses. Furthermore, low-energy barrier translational movements are also consistent with the reversible nature of the transition from immature trimer to dimer observed previously[13]. Finally, we suggest that the immature conformation of the prM-E complex is under strain and protonation of His101 of the prM linker (SPOV numbering) in the trans-Golgi leads to release of E, allowing relaxation into the flat-lying conformation. A topologically feasible, tentative model of the immature-mature reorganization is presented in Supplementary Movie 2.

Early work on flavivirus maturation has indicated that furin cleavage of prM does not occur at neutral pH[11] (as in the ER or early Golgi), when the virus is in the spiky conformation. This led to the suggestion that the protease recognition site is not exposed. Later modeling based on low resolution data inferred that the furin recognition motif is indeed sterically blocked in the immature conformation[31]. In the current study, our map resolution allows us to elucidate the molecular basis of furin site occlusion within the prM$_3$E$_3$ trimer. However, furin protease is inactivated by binding its prosegment before reaching the low pH of the TGN, where the inhibitory prosegment is released[14]. If furin is inactive in the ER and early Golgi this begs the question why it is necessary for the virus to obscure the furin site in these early compartments. However, previous studies have shown that furin is occasionally responsible for some proteolytic processing in the ER or early Golgi, for instance, if the substrate outcompetes the inhibitory prosegment[37,38]. We suggest that occlusion of the furin site within immature trimers is a critical adaptation for the virus because otherwise high levels of virus expression within an infected cell, combined with high local concentrations of furin recognition motifs on a viral surface could lead to some cleavage events in the ER or early Golgi. As the pH is neutral in these compartments, the affinity of cleaved pr to E would be low and some pr could dissociate. In turn, this would lead to partially matured virus (which is fusion competent[3]) reaching the low-pH TGN, where unproductive fusion would be triggered prematurely.

We have shown that, in addition to Sindbis alphavirus[35], DENV2, ZIKV, and SPOV all possess a lipid-like density clenched between the membrane-associated helices of their glycoproteins. The high-resolution map for SPOV allows us to assign the density to a PE type lipid. Previous computational modeling studies, that took lipidomics into account[29], have estimated that there are around 2400 PE lipids in the dengue virion. Tight binding of 180 PE molecules by all copies of E would constitute the immobilization of 7.5% of total PE in the virion, a substantial amount. This lipid site may be involved in the stabilization of the mature virus, preventing the space between membrane-associated helices from collapsing and ensuring their correct orientations. Given the observation of a structurally equivalent lipid in alphaviruses, it is tempting to speculate that this is a strategy common among enveloped viruses. Furthermore, a highly conserved and mutation-sensitive histidine (His447 in SPOV, His437 in DENV2) is located adjacent to the lipid, suggesting a pH-dependent role during the viral lifecycle. Just as small molecules mimicking a lipid pocket factor in picornaviruses have similarly been successfully employed as viral inhibitors[39–41], we suggest that the pocket may be a potential target for the development of antivirals.

## Methods

**Virus sample preparation and inactivation**. SPOV strain SM6 V-1s and DENV2 strain 16681 were propagated in C6/36 cells (gift from Dr. Malasit, Mahidol University, Thailand) maintained in L15 (Thermo Fisher Scientific) supplemented with 1.5% heat-inactivated fetal calf serum, 1 mM glutamine, and penicillin/streptomycin. To prepare virus for cryo-EM analysis, cell-free virus supernatants were concentrated by polyethylene glycol (PEG) precipitation using PEG-8000, to a final concentration of 8%. The supernatant/PEG mixture was stored overnight at 4 °C before centrifugation at 3200 × g for 90 min at 4 °C.

Pelleted viral particles were resuspended in cold NTE buffer (12 mM Tris, pH 8.0, 120 mM NaCl, 1 mM EDTA), cleared by centrifugation at 3200 × g, 5 min, 4 °C, layered over a 22% sucrose cushion and centrifuged at 175,000 × g (Beckman Coulter Sw41) for 2 h at 4 °C. The pellet was resuspended in cold NTE buffer and kept overnight at 4 °C and was subsequently applied to a 10–35% potassium tartrate step gradient. The virus-containing fraction was collected and buffer exchanged into NTE buffer by multiple rounds of dilution and centrifugation using a 100 kD cutoff centrifugal filter device (Amicon Ultra).

To identify optimal UV-exposure conditions for inactivation, virus samples were placed into a UV crosslinker (Uvitec) for different periods of time (from 15 s to 5 min). UV-inactivated virus samples were added to a Vero cell monolayer and incubated for 2 h. Non-UV-inactivated virus was used as a control. A plaque assay was performed by topping up the cells with 1.5% carboxymethyl cellulose and incubation at 37 °C for 3 days to allow virus infection. Finally, viral foci were visualized via the cross-reactive mouse monoclonal antibody 4G2 (gift from Dr Malasit, Mahidol University, Thailand), followed by anti-mouse immunoglobulin conjugated to horseradish peroxidase and development by DAB (3,3'Diaminobenzidine) substrate. Although we saw complete inactivation after the minimal exposure of 1 min, we doubled this inactivation time to 2 min for electron microscopy samples. Following inactivation, a volume of 3.5 µl of purified SPOV was pipetted on glow-discharged Quantifoil holey carbon grids (1 µm spacing, 2 µm holes, biocompatible 200 gold mesh) and was blotted for 3.5 s before flash-freezing in liquid ethane using a Vitrobot mark IV (FEI). For DENV, Protochips carbon-coated copper C-flat grids (1 µm spacing, 2 µm holes) were used. A sample of the quality of UV-inactivated virus is shown in Supplementary Fig. 1b.

**Cryo-EM data collection and processing**. Cryo-EM movies for SPOV were collected at the Electron Bio-Imaging Centre (eBIC) at Diamond Light Source, UK. A 300 kV Titan Krios instrument (Thermo Fisher Scientific), equipped with a K3 (Gatan) direct detector and a GIF Quantum energy filter (Gatan) with 20 eV slit width was utilized. Data were collected automatically using SerialEM 3.7 in super-resolution mode (0.4225 Å per pixel). DENV2 data were collected at the Division of Structural Biology, Oxford, UK, on a 300 kV F30 Polara (Thermo Fisher Scientific) equipped with a K2 Summit (Gatan) direct detector (super-resolution mode, 0.44 Å per pixel) and a GIF Quantum energy filter (Gatan) with 20 eV slit width. Data collection parameters are summarized in Table 1.

Motion correction of cryo-EM movies was carried out using MotionCor2-1.1.0[42], and 2× binning was applied for super-resolution data. Contrast transfer function (CTF) parameters were estimated using Gctf-v.1.06[43] (SPOV) and CTFFIND4[44] (DENV2). Particles were picked using cryoSPARC v.2.12.0[45] (SPOV) and Ethan1.2[46] (DENV2). Subsequent classification and refinement steps were performed in RELION3.1[47]. Icosahedral reconstruction (I1 symmetry) of immature SPOV and mature SPOV/DENV2 was carried out using standard methods. In brief, the initial particle sets were cleaned via 2D/3D classifications runs, followed by masked 3D refinement and post-processing. CTF refinement and Bayesian polishing in RELION3.1 were used to further improve the density and resolution of the reconstructions. The numbers of initial and final particles are summarized for all samples in Table 1.

To acquire high-resolution maps of immature spikes, we used Localized Reconstruction[23] within Scipion v.2.0[48,49] to obtain coordinates, CTF parameters, and orientations of individual spike subparticles and carried out processing of subparticles in RELION. An overview of the workflow is shown in Supplementary Fig. 2. In brief, we extracted subparticles of immature trimeric spikes from 42,880 viruses. The resulting 2,572,800 subparticles were then subjected to 3D classification without alignment. The 3D classes revealed a substantial amount of heterogeneity in the spikes (see Supplementary Fig. 2). This underscores why icosahedral reconstructions of immature flaviviruses would be limited in resolution, as heterogeneous spikes would be averaged when applying symmetry. In addition, virus preparations typically contain partially immature particles[6], and mature patches on such hybrid particles may also contribute to heterogeneity. We selected two classes with a total of 371,915 subparticles for further processing. A second round of classification, followed by masked refinement and post-processing, yielded a 4.2 Å resolution map of the prM$_3$E$_3$ trimeric spike reconstructed from 305,017 subparticles. Although the quality of the density had dramatically improved by localized reconstruction, we attempted to improve the resolution further to aid in model building and refinement. We performed focused classification and refinement on prM$_1$E$_1$ monomers and finally obtained a map at 3.8 Å resolution, reconstructed from 281,619 subparticles. This map served as the basis for initial building of an atomic model, while the 4.2 Å resolution map of prM$_3$E$_3$ was used to refine an atomic model of the full trimer. Local resolution values of the immature localized reconstructions and map vs model FSC plots are shown in Supplementary Fig. 3.

All atomic structures in this study were refined by alternating cycles of real-space refinement in PHENIX v.1.17.1[50] and manual building in Coot v.0.8.9.2[51]. Model geometry was validated using MolProbity v.4.2[52]. Geometry statistics, model B factors, and map vs model cross-correlation values are shown in Table 1 for all refined models.

**Construction of mutant viruses**. To introduce amino-acid exchanges at the lipid pocket, site-directed mutagenesis of prM/E in a pHLsec vector[53] was performed

using Pfx DNA polymerase (Invitrogen) according to the manufacturer's protocol. Primer sequences are listed in Supplementary Table 1.

To produce virus from DNA, five PCR products for assembly of viral constructs were amplified with Phusion® High-Fidelity DNA polymerase (NEB) according to the manufacturer's protocol. PCR product of the expression vector was amplified with pcDNA-hCMV-R and DV1-2-3-3UTR-HDVAr-F. PCR products of cDNAs of DENV2-16681 genomes were carried out by three pairs of primers: hCMV-DV2-16681-5UTR-F and DV2-16681prM-start-R (F1), DV2-16681-Eend-F and DV2-16681-6685-6709-R (F3), and DV2-16681-6695-6724-F and DV1-2-3-3UTR-R (F4). PCR product of the pHLsec-prM/E wild-type and mutants were amplified by DV2-16681prM-start-F and DV2-16681-Eend-R (F2). All PCR products were purified using the QIAquick Gel Extraction Kit (QIAGEN).

PCR products of the expression vector and viral PCR products were then assembled using the Gibson assembly technique[36,54]. PCR products of the expression vector (0.04 pmol) were assembled with 0.04 pmol of each PCR product of the viral genome in the enzyme mix consisting of 5× isothermal buffer (25% PEG-8000, 500 mM Tris-HCl pH 7.5, 50 mM MgCl$_2$, 50 mM DTT, 1 mM each of the four dNTPs and 5 mM NAD), T5 exonuclease, Taq DNA ligase and Phusion DNA polymerase (New England Biolabs). The reaction was incubated at 50 °C for 3 h.

Twenty microliters of the Gibson assembly reactions were diluted in Opti-Minimal Essential Media I (Thermo Fisher Scientific), mixed with Lipofectamin 2000 (Invitrogen), before transfecting human embryonic kidney 293 T cells (HEK-293T, ATCC, CRL-3216) for 4 h at 37 °C. The medium was then exchanged to Dulbecco's Modified Eagle Medium (Thermo Fisher Scientific), supplemented with 10% heat-inactivated fetal calf serum. After overnight incubation, the medium was removed and replaced by Ultradoma Protein-Free (LONZA). The cultured medium was harvested and replenished on the second, third, and fourth days after transfection. Cell-free supernatants were collected and stored at −80 °C. Wild-type and mutant viruses from transfected HEK-293T cells were further propagated in C6/36 cells. Medium was changed on the fourth day and then every 3 days. Viral production and titers were determined by enzyme-linked immunosorbent assay and focus-forming assay, as described previously[55]. In brief, to determine DENV production, virus supernatants were captured onto plates coated with antibody 4G2 and then incubated with antibody 749B12 (human anti-fusion loop epitope, previously reported by Dejnirattisai et al.[55]), followed by alkaline phosphotase-conjugated anti-human IgG (A9544; Sigma). Reactions were developed by the addition of para-nitrophenylphosphate substrate and were stopped by addition of NaOH. The absorbance was measured at 405 nm.

**Reporting summary.** Further information on research design is available in the Nature Research Reporting Summary linked to this article.

## Data availability

All data are available from the corresponding authors and/or are included in the manuscript. Cryo-EM density maps and atomic coordinates of models have been deposited in the Electron Microscopy Data Bank and the Protein Data Bank (PDB), respectively. Atomic coordinates are deposited under the PDB accession codes PDB-ID 6ZQU (mature DENV2, icosahedral reconstruction), PDB-ID 6ZQV (mature SPOV, icosahedral reconstruction), PDB-ID 6ZQJ (immature SPOV trimeric spike, localized reconstruction), PDB-ID 6ZQI (immature prM$_1$E$_1$ monomer, localized reconstruction), and PDB-ID 6ZQW (immature SPOV, icosahedral reconstruction). Cryo-EM maps are deposited under the EMD accession codes EMD-11370 (mature DENV2, icosahedral reconstruction), EMD-11371 (mature SPOV, icosahedral reconstruction), EMD-11366 (immature SPOV trimeric spike, localized reconstruction), EMD-11364 (immature prM$_1$E$_1$ heterodimer, localized reconstruction), and EMD-11372 (immature SPOV, icosahedral reconstruction). Source data are provided with this paper.

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

## Acknowledgements

This work was supported by the Wellcome Trust, UK. Microscopy was partly conducted at the OPIC electron microscopy facility, which was funded by a Wellcome JIF award (060208/Z/00/Z) and is supported by a Wellcome equipment grant (093305/Z/10/Z). The Wellcome Trust is also acknowledged for providing administrative support (Grant 075491/Z/04). GRS is a Wellcome Trust senior investigator. MR is supported by a Wellcome Trust fellowship (204703/Z/16/Z). JMG is supported by a Wellcome Investigator Award (200835/Z/16/Z). ISM was supported by an MRC studentship (1374922). We thank B. Qureshi for support with electron microscopy, the staff of the Electron Bio-Imaging Centre (eBIC), especially D. Clare, for assistance with data collection and advice on data processing. We also thank L. Mendonça, T. Frosio, and members of the P. Zhang group for scientific discussions. Computation used the Oxford Biomedical Research Computing (BMRC) facility, a joint development between the Wellcome Centre for Human Genetics and the Big Data Institute, supported by Health Data Research UK and the NIHR Oxford Biomedical Research Centre. Financial support was provided by a Wellcome Trust Core Award (203141/Z/16/Z). The views expressed are those of the author(s) and not necessarily those of the NHS, the NIHR, or the Department of Health.

## Author contributions

M.R., W.D., J.M., G.R.S., and J.M.G. conceived and designed the study. W.D. and J.M. prepared and purified virus samples and performed Gibson assembly. M.R., L.C., and I.S.M. prepared vitrified cryo-EM grids. M.R., L.C., D.K., and J.K. screened grids. M.R., L.C., D.K., S.F.H., and A.K. collected cryo-EM data. M.R. carried out cryo-EM data processing and model building/refinement with assistance from L.C., I.S.M., and S.L.I. M.R. wrote the manuscript, with input from all the authors.

## Competing interests

The authors declare no competing interests.
