## [Peer Review File · Nature Communications]

Reviewers' Comments:

Reviewer #1:

Remarks to the Author:

The manuscript by Renner et.al., describes a high resolution structure of immature SPOV, a flavivirus similar to other prominent members of the family such as ZIKV and Dengue viruses. They have used localized reconstruction of the prM-E trimers and prM-E monomers to push the resolution of the structure to 4.2Å and 3.8Å respectively. Their manuscript also reports a 2.6Å and 3.1Å structure of SPOV and DENV particles. Although the high resolution structures are commendable and the accompanied structural details observed are a useful addition to our knowledge of flavivirus structures, the manuscript overall does not deliver any significant breakthrough on the flavivirus structures which was not known before.

The occlusion of the furin cleavage site and its position in the prM-E structure has been known from previous work by Li et.al and Yu et.al., Science,2008. The current work confirms much of this information by tracing these regions in full virus at higher resolution. Cryo-EM structures of immature dengue virus (Kostyuchenko et.al.) and immature zika virus (Mangala Prasad et.al.,) have previously shown the differences in arrangement of transmembrane (TM) domains of M and E when compared to their mature counterparts. This work again confirms much of this information. They also identify a lipid binding pocket between the TM domains in the mature SPOV which has been recently also described for ZIKV in DiNunno et.al.,2020.

The authors propose a different model for structural transition between the immature and mature structures. Although the proposed model requires less movement between the different proteins compared to previously proposed models, it is still speculative and solely based on endpoint structures. Overall, the manuscript is a well written research work which further underlines the remarkable structural similarity within flaviviruses.

Reviewer #2:

Remarks to the Author:

In this paper, Max Renner et. al., reconstructed immature Flavivirus at ~3.9 Å resolution, which enables the building of an atomic model. The structure shows how the furin cleavage site being buried and how E interacting with pr. In addition, a conserved histidine is identified to anchor the pr. A comparison of immature and mature Flaviviruses near the viral membrane reveals a lipid binding pocket only existed in the mature virus. The mutagenesis experiment indicates this pocket is important in the virus lifecycle.

The structural data is of good quality and supports the identified protein interactions. The findings of detailed structure of immature virus is important for the understanding of the assembly of immature virus and the structure also gave the hint of the structure rearrangement during maturation. Although, the finding of lipid-binding hydrophobic pocket is similar to other paper that published recently. The earlier version of this manuscript has been submitted to bioRxiv for a long time. Overall, the manuscript is of good quality and novelty and is worthy to be published. I only have some minor points.

Page 5, However, the last clearly visible residue prior to this flexible region is a histidine (His101), which is buried in a hydrophobic pocket on E. Do you mean a Histidine that is inserted into a hydrophobic pocket?

Page 8, no intermediate structures available. A low resolution structure of immature dengue virus has been determined at pH 6.0 by I-Mei Yu et. al., which is Ref. 12 in the manuscript.

At low pH, the E protein may dissociate from M and forms pre-fusion or post-fusion trimer on virion (Journal of virology 89 (1), 743-750). However, at low pH, the E protein of immature virus tends to

form dimer which is similar to that in the mature virus. Is it possible to discuss which unique interactions in immature virus may remain in the low pH and may cause this difference? The lack of lipid-binding pocket? The interaction between E and pr? Why is the flat-lying conformation preferred in the trans-Golgi for immature virus while a more spiky conformation is preferred of mature virion at low pH? The pH dependent conformation change of immature virus is reversible. Can the "translation" model explain the reversibility? It may worth to discuss above questions in the discussion.

Xinzheng Zhang

Response to Reviewers

We thank the reviewers for their encouraging comments and we are grateful for the opportunity to revise our manuscript. We believe that the revised manuscript is strengthened, such that it states the scientific advance more clearly and better places it into the context of the field. Please find below point-by-point responses to the raised issues.

Reviewer #1 (Remarks to the Author):

The manuscript by Renner et.al., describes a high resolution structure of immature SPOV, a flavivirus similar to other prominent members of the family such as ZIKV and Dengue viruses. They have used localized reconstruction of the prM-E trimers and prM-E monomers to push the resolution of the structure to 4.2Å and 3.8Å respectively. Their manuscript also reports a 2.6Å and 3.1Å structure of SPOV and DENV particles. Although the high resolution structures are commendable and the accompanied structural details observed are a useful addition to our knowledge of flavivirus structures, the manuscript overall does not deliver any significant breakthrough on the flavivirus structures which was not known before.

The occlusion of the furin cleavage site and its position in the prM-E structure has been known from previous work by Li et.al and Yu et.al., Science,2008. The current work confirms much of this information by tracing these regions in full virus at higher resolution.

We agree with the reviewer that we may not have stated clearly enough the degree to which our higher resolution structure improves upon what was known previously. To the best of our knowledge, our structure is the first refined atomic model of any immature flavivirus, including the critical furin protease site. To put this into the context of the literature, the studies by Li et al., Yu et al., and Mangala Prasad et al., mentioned by the reviewer, were limited to rigid body fitting of crystal data into the immature virus map, with no models deposited for the furin site. In the former study, as far as we are able to tell, the authors employed rough tracing to identify an approximate area where the furin site might plausibly be located, based on chain length. However, at low resolution there is great uncertainty about the real position of the site as we will illustrate further below.

Even the most recent and state-of-the-art study on the structure of the immature ZIKV (Tan et al., 2020, PMID: 32060358), reported at $\sim 8\text{\AA}$, is limited to a C-alpha trace with no sidechain information. Indeed, when comparing our refined atomic model with the 8\AA C-alpha trace model we observe large discrepancies in the main chain trace (see Figure below), which are not expected given the $\sim 70\%$ sequence identity. These are likely due to the inherent uncertainty in a 8\AA map. Notably, the position of the furin site in the 8\AA vs our 3.8\AA structure is out of register by 2.5 residues. Given the structural similarity between flaviviruses in general, and SPOV/ZIKV specifically, the furin sites will be in the same position. Our work builds upon the important previous studies, and emphasises how our higher resolution data provides more critical detail. In particular, for researchers in the field designing mutants for functional experiments, an accurate model is crucial to be able to target the correct interactions (e.g. interactions between the furin site and E). To summarize, as the first refined atomic model of an immature flavivirus spike, we believe our structure constitutes a significant advance that goes far beyond confirming the previous model and will be of great importance to the flavivirus community in providing an accurate framework for structure-function experiments in all related flaviviruses. Furthermore, with our localized

reconstruction approach, we provide a roadmap for the high resolution structure determination of other flexible immature viruses in the future. Finally, we were also the first to report the ordered lipid (detailed below), underlining the impact of our study. We have revised the abstract and the text to more strongly emphasize how our study constitutes a significant advance relative to the field.

Fig. 1. Comparison of cryo-EM structures of SPOV prM-E at 3.8Å (this work) and ZIKV prM-E at ~8Å (pdbid: 6LNU). SPOV and ZIKV structures are colored in white and grey, respectively. The positions of the modelled furin RSRR motifs are highlighted in blue (SPOV) and red (ZIKV).

Cryo-EM structures of immature dengue virus (Kostyuchenko et.al.) and immature zika virus (Mangala Prasad et.al.) have previously shown the differences in arrangement of transmembrane (TM) domains of M and E when compared to their mature counterparts. This work again confirms much of this information. They also identify a lipid binding pocket between the TM domains in the mature SPOV which has been recently also described for ZIKV in DiNunno et.al.,2020.

As correctly alluded to by Reviewer #2 our biorxiv preprint (<https://doi.org/10.1101/2020.06.07.138669>) was the first work to report the lipid binding pocket. It has been available since June 2020, thus preceding the paper from DiNunno et al (published in October 2020). Indeed, our preprint was released and publicly available (June 07, 2020) before the DiNunno paper was submitted to Nature Communications (25 June, 2020). The authors even cited our preprint in their paper. Furthermore, in our preprint we already showed the presence of the ordered lipid in not only SPOV, but also in ZIKV and DENV, although DiNunno et al., did not mention this when referencing us. We have now added a sentence in our manuscript citing the paper of DiNunno et al., and clarifying the sequence of findings.

The authors propose a different model for structural transition between the immature and mature structures. Although the proposed model requires less movement between the

different proteins compared to previously proposed models, it is still speculative and solely based on endpoint structures. Overall, the manuscript is a well written research work which further underlines the remarkable structural similarity within flaviviruses.

We agree that the transition model is speculative (as indeed was the previous model), and this is why we propose it at the very end of the text. However, we do think it is a good starting point for further investigations and believe it benefits from the fact that we can model a clash-free transition from a complete spikey to a complete smooth virus (rather than just modelling a transition in a single asymmetric unit, which may obfuscate serious clashes in the full virus), as shown in the supplementary movie 2.

Reviewer #2 (Remarks to the Author):

In this paper, Max Renner et. al., reconstructed immature Flavivirus at ~3.9 Å resolution, which enables the building of an atomic model. The structure shows how the furin cleavage site being buried and how E interacting with pr. In addition, a conserved histidine is identified to anchor the pr. A comparison of immature and mature Flaviviruses near the viral membrane reveals a lipid binding pocket only existed in the mature virus. The mutagenesis experiment indicates this pocket is important in the virus lifecycle.

The structural data is of good quality and supports the identified protein interactions. The findings of detailed structure of immature virus is important for the understanding of the assembly of immature virus and the structure also gave the hint of the structure rearrangement during maturation. Although, the finding of lipid-binding hydrophobic pocket is similar to other paper that published recently. The earlier version of this manuscript has been submitted to bioRxiv for a long time. Overall, the manuscript is of good quality and novelty and is worthy to be published. I only have some minor points.

We thank the reviewer for their positive comments.

Page 5, However, the last clearly visible residue prior to this flexible region is a histidine (His101), which is buried in a hydrophobic pocket on E. Do you mean a Histidine that is inserted into a hydrophobic pocket?

Yes, we have modified the text to state this more clearly.

Page 8, no intermediate structures available. A low resolution structure of immature dengue virus has been determined at pH 6.0 by I-Mei Yu et. al., which is Ref. 12 in the manuscript.

Indeed, there is a low resolution map available of a low pH virus with uncleaved prM. As the reviewer correctly states below, this virus likely features dimeric E protein, arranged in a herringbone fashion. In our manuscript what we meant to say was that there is no intermediate between a dimeric arrangement of E and the trimeric spike arrangement, which leaves the trajectory of this transition underdetermined. We have now clarified this in the text.

At low pH, the E protein may dissociate from M and forms pre-fusion or post-fusion trimer on virion (Journal of virology 89 (1), 743-750). However, at low pH, the E protein of

immature virus tends to form dimer which is similar to that in the mature virus. Is it possible to discuss which unique interactions in immature virus may remain in the low pH and may cause this difference? The lack of lipid-binding pocket? The interaction between E and pr? Why is the flat-lying conformation preferred in the trans-Golgi for immature virus while a more spiky conformation is preferred of mature virion at low pH?

We thank the reviewer for this excellent question and we think that a serious limitation to answering it is the absence of a high resolution structure of the immature low-pH virus. The available structure (pdbid 3C6R) is based on rigid body fitting, and is missing all transmembrane helices and, most importantly, the pr-M linker. A good structure of the low-pH immature virus would show if the ordered lipid is incorporated at this stage. This is a very interesting question, because the low pH would change the protonation of His101 and thus its interaction with the lipid. It would also show if there are movements in the TM-helices, how the pr-M linker is threaded in between the E-dimer, and most importantly, how the furin site is exposed. We think that this would also greatly help in carefully analysing the interactions between pr and E, and lead to a model of how the presence of pr (cleaved or uncleaved) determines the state of the virus at low pH. Before such a structure is solved we would prefer to not speculate on this too much.

The pH dependent conformation change of immature virus is reversible. Can the “translation” model explain the reversibility? It may worth to discuss above questions in the discussion.

This is a very good point and we do believe that in our proposed model the observed reversibility is more accounted for than in a model in which there are large scale rotations of ectodomains. As we suggest that the transition occurs predominantly by diffusion in the fluid lipid membrane (“sliding of proteins”), with smaller-scale conformation changes, the expected energy barrier may be lower than for a transition involving large scale rotations. A low energy barrier is consistent with an appreciable reverse reaction rate and facilitates backwards transition under the altered pH conditions. We now note this in the discussion.